# Electronic States of Epigallocatechin-3-Gallate in Water and in 1,2-dipalmitoyl-sn-glycero-3-phospho-(1′-rac-glycerol) (Sodium Salt) Liposomes

**DOI:** 10.3390/ijms26031084

**Published:** 2025-01-27

**Authors:** Filipa Pires, Demeter Tzeli, Nykola C. Jones, Søren V. Hoffmann, Maria Raposo

**Affiliations:** 1Laboratory of Instrumentation, Biomedical Engineering and Radiation Physics (LIBPhys-UNL), Department of Physics, NOVA School of Science and Technology, Universidade NOVA de Lisboa, 2829-516 Caparica, Portugal; ana.s.pires@ist.utl.pt; 2Instituto de Telecomunicações, Instituto Superior Técnico, Universidade de Lisboa, Av. Rovisco Pais, 1049-001 Lisboa, Portugal; 3Laboratory of Physical Chemistry, Department of Chemistry, National and Kapodistrian University of Athens, Panepistimiopolis Zografou, 157 84 Athens, Greece; 4Theoretical & Physical Chemistry Institute, National Hellenic Research Foundation, 48 Vassileos Constantinou Ave., 116 35 Athens, Greece; 5ISA, Department of Physics and Astronomy, Aarhus University, Ny Munkegade 120, 8000 Aarhus C, Denmark; nykj@phys.au.dk (N.C.J.); vronning@phys.au.dk (S.V.H.)

**Keywords:** EGCG, epigallocatechin-3-gallate, 1,2-dipalmitoyl-sn-glycero-3-phospho-(1′-rac-glycerol) (sodium salt), DPPG, VUV, electronic transitions, water, liposome, molecular interaction

## Abstract

In this work, the spectroscopy of epigallocatechin-3-gallate (EGCG) and EGCG bonded to 1,2-dipalmitoyl-sn-glycero-3-phospho-(1′-rac-glycerol) (sodium salt) (DPPG) lipid is studied both experimentally by combining high-resolution vacuum ultraviolet (VUV) photo-absorption measurements in the 4.0–9.0 eV energy range and by theoretical calculations using density functional theory (DFT) methodology. There is a good agreement between the experimental and theoretical data, and the inclusion of the solvent both implicitly and explicitly further improves this agreement. For all experimentally measured absorption bands observed in the VUV spectra of EGCG in water, assignments to the calculated electronic transitions are provided. The calculations reveal that the spectrum of DPPG-EGCG has an intense peak around 150 nm, which is in accordance with experimental data, and it is assigned to an electron transfer transition from resorcinol–pyrogallol groups to different smaller groups of the EGCG molecule. Finally, the increase in absorbance observed experimentally in the DPPG-EGCG spectrum can be associated with the interaction between the molecules.

## 1. Introduction

Natural anti-inflammatory and antioxidant molecules, such as catechins, prevent and slow down the cascade of oxidative events in biomolecules (such as DNA, phospholipids, and proteins) caused by oxidative stress [1,2,3,4,5,6,7,8,9,10]. Considering this, catechins have been integrated into various scaffolds in tissue engineering and regenerative medicine to expedite healing [11,12,13,14], promote tissue regeneration [15,16,17,18,19,20], and modulate angiogenesis [21,22,23,24,25].

The amount of catechins like Epigallocatechin-3-gallate (EGCG) in one cup of green tea is enough to inhibit the reactivity of reactive oxygen species (ROS) inside the body, thereby reducing oxidation-induced lesions [26,27,28,29]. Moreover, EGCG could reduce side effects stemming from radiation exposure in human dermal fibroblast cells [30,31,32,33,34,35,36,37,38].

Our previous studies revealed that the incorporation of EGCG protects the 1,2-dipalmitoyl-sn-glycero-3-phospho-(1′-rac-glycerol) (sodium salt) (DPPG) membrane, namely the phosphate and carbonyl groups of phospholipids, from oxidative stress induced by ultraviolet (UV) radiation [39,40,41]. Therefore, it is crucial determine the maximum amount of EGCG that can be immobilized into phospholipid membranes and reveal the chemical reactivity underlying its antioxidant activity so that these molecules could further be implemented in food packing systems, cosmetics, and cancer therapeutics.

The bioactivity of catechins depends on its structural and molecular properties, which can be determined by spectroscopic experiments combined with theoretical calculations. For instance, Antonczack and co-workers [42] obtained catechin dipole moment values of 3.22 D and 4.05 D associated with the rotation of the catechol moiety around the C2C1’bond using the B3LYP/6-31++G(d,p) density functional method. The molecular simulations for catechin showed that its HOMO and LUMO orbitals are localized either on the AC-conjugated ring (Figure 1) or on the catechol moiety and that electron transfer does not occur from AC rings towards the B ring under the deprotonation of the O3H and O4′H hydroxyl groups. Regarding the hydrogen abstraction process, simulations have shown that it causes small deformations on each structural motif and leads to the formation of four stable radicals: 4′−O^●^, 3′−O^●^, 5−O^●^, and 7−O^●^ for catechin [43]. These molecular simulations reveal that catechins use two essential mechanisms for free radical scavenging: the single electron transfer (SET) and the hydrogen atom transfer (HAT). The antioxidant efficiency of each type of catechin involving the HAT mechanism is correlated with the bond dissociation enthalpy (BDE) of the phenolic O−H bond, meaning that catechins with lower BDE values are easily deprotonated and, thus, much more effective in radical scavenging. For instance, molecular simulations estimated a BDE value of 66.46 kcal/mol for the hydroxyl group at position 5′ in ring B of catechin, making it the most reactive aromatic ring during the elimination of excessive ROS [44]. Moreover, the differences in scavenger behavior of catechin stereoisomers, like (+)-catechin and (−)-epicatechin, arise due to the variations in the intrinsic chemical reactivity and the position–orientations of the B ring concerning the C ring [45,46]. Indeed, the sites for electrophilic attack and electron charge distribution on hydroxyl groups attached to rings are different in these molecules, thus generating different intermediate oxidation products (e.g., semiquinone and quinone products).

Despite the presence of phenolic hydroxyl groups increasing the antioxidant/scavenging activity of catechins, it is known that the oxidation of the catechol group on the B ring of catechin produces catechins by-products (quinone derivatives) that can cause adverse effects on biomolecules. According to electrochemical measurements, catechin’s oxidation is a pH-dependent mechanism, where the catechol B ring is more easily oxidizable because it has a lower redox potential [47]. At very low positive potentials, (+)-catechin has a high radical scavenging activity, because there occurs a reversible oxidation in its catechol moiety. This scavenging activity increases at neutral pH due to the deprotonation of the trihydroxyl group in the B moiety (3′4′5′-OH), which facilitates electron transfer [48]. It is expected that, besides the pH, other parameters such as the solvent type and temperature influence the antioxidant and ROS scavenger properties of catechins. For instance, the position of the maximum absorbance peak of EGCG is centered at 273 nm in water but shifts towards higher wavelengths in methanol (276 nm). Moreover, the effect of the solvent environment on the spectroscopic properties of gallic acid [49,50] and dihydro-xybenzene [51,52,53], which are parts of EGCG, have been also studied both theoretically and experimentally [49,50,51,52,53,54,55].

Even though there are many studies on EGCG and its components, as far as we know, the characterization of the low-lying electronic and ionic states of catechins has not been conducted, namely in the high-resolution vacuum ultraviolet (VUV) energy range. This knowledge is relevant to assess the role of certain electronic states participating in key antioxidant reaction mechanisms.

In the present work, we used both experimental VUV photo-absorption measurements, in the 4.0–9.0 eV energy range, and molecular simulations using the DFT methodology, to characterize the electronic transitions of the EGCG molecule in a water solvent. Furthermore, the electronic transitions of EGCG were also analyzed upon its interaction with DPPG phospholipids, considering the biological role of this molecule in protecting cellular membranes against UV radiation-induced damage.

## 2. Results and Discussion

### 2.1. VUV Characterization of EGCG

Figure 2a shows the VUV spectrum of EGCG cast thin films prepared from aqueous solutions. This spectrum reveals seven bands centered at the following wavelengths: 140, 159, 172, 212, 228, 272, and 292 nm. Table 1 shows the positions of the absorption bands detected in the VUV spectrum of an EGCG cast film as well as the respective FWHM and the assignments of the electronic transitions involved; see discussion below.

### 2.2. Calculations of EGCG in Water

#### 2.2.1. Conformational Analysis and Energetics

DFT and TD-DFT calculations were carried out using the B3LYP, M06-2X, PBE0, ωB97XD functionals and the MP2 perturbation theory in conjunction with the 6-31G(d,p) and 6-311+G(d,p) basis sets to study the geometry and the electronic structure of EGCG in a water solvent. First, a conformational analysis was carried out to find the lowest energy minimum structures of EGCG. Eight stable minima of EGCG were obtained, as shown in Figure 3a–h. The solvent was included implicitly and explicitly with one, seven, and eleven water molecules, as shown in Figure 4. Eleven water molecules were added because the EGCG molecule has eleven oxygen atoms, while seven water molecules were added because, after the interaction of lipids with catechin, seven oxygen atoms were free to form van der Waals bonds with the water molecules, as shown in Figure 4. In addition, the interaction of the DPPG lipid with EGCG was calculated via the ONIOM method in a water solvent with and without the implicit inclusion of water molecules; see Section 3: Materials and Methods. The energetics and selected geometries of the minimum structures are presented in Table 2. Additionally, the interaction energy between water molecules and EGCG was calculated, and the corrected values for the basis set superposition error (BSSE) are also provided in Table 2.

The eight stable isomers of the EGCG molecule can be split into two groups: the first contains the **a**–**d** isomers and the second the **e**–**h** isomers, where a van der Waals bond H…O between the H of the pyrogallol ring and O of the gallate ring of about 1.98 Å is formed, as shown in Figure 3. The isomers differ in the relative positions of the H atoms of the OH groups. Note that both the pyrogallol and the resorcinol rings rotate with respect to the gallate ring when this type of van der Waals interaction is formed, as shown in Figure 3. These rotations result in an increase in the relative energy of about 0.5 kcal/mol. These eight isomers are lying energetically within 3.4 kcal/mol in water; see Table 2. Note that, in the gas phase, they are all within 6.0 kcal/mol and the relative ordering is different within each of the two groups. However, in both cases, the most stable isomer is the same minimum. Additionally, all used methodologies predicted the same geometry. The dihedral angles between the resorcinol, gallate, and pyrogallol rings in water, dRG, dRP, and dGP are about 117, 92, and 38 degrees for the **a**–**d** isomers and change to 155, 98, and 60 degrees for the **e**–**h** isomers where the H…O bonds are formed, respectively. Note that the dRP dihedral angle is similar in both groups of the isomers.

The interaction of one water molecule with EGCG (see EGCG_w, Figure 4) was calculated at −6.15 (−5.35) [−4.65] in a water solvent via the ωΒ97ΧD (MP2) [PBE0] method in conjunction with the 6-311+G(d,p) basis set. The BSSE error is about 1 kcal/mol, i.e., −5.05 (−4.29) [−3.63] kcal/mol. The addition of the dispersion corrections increases the calculated values of the interaction energy. Two O…H bonds are formed between the water molecule and EGCG, and their calculated bond distances are 2.233 [2.284] Å and 2.073 [2.071] Å with the ωΒ97ΧD/6-311+G(d,p) [PBE0/6-311+G(d,p)] methods, showing that both functionals predict similar O…H bonds.

The interaction energy between EGCG and the DPPG lipid was calculated at −8.9[8.5] kcal/mol, while between EGCG and the DPPG lipid + 7 water molecules was calculated at −9.6[9.4] kcal/mol at the PM6:PBE0/6-31G(d,p)[PM6:PBE0/6-311+G(d,p)] levels of theory; see the minima structures in Figure 4.

#### 2.2.2. Calculated VUV Absorption Spectra of EGCG

Comparing the different functionals used, we observe that they present the same general shape of the absorption spectrum of the EGCG. However, there are some differences, as shown in Table 3. The B3LYP and PBE0 functionals are in close agreement with the entire B3LYP spectrum moved to lower energies by about 0.3 eV, while the M06-2X spectrum presents some differences. This relative behavior of the three functionals used has been observed in other molecules; see, for instance, reference [58]. Comparing the absorption spectrum in the gas phase and in a water solvent, for all functionals, the peaks are slightly shifted in the solvent by about 10 nm (about 0.1–0.2 eV) and, in most cases, to lower energies. Finally, comparing the two basis sets 6-31G(d,p) and 6-311+G(d,p), the large basis set presents peaks shifted to lower energies. While the shifts are not large in the case of the lower energy transitions, in the high-energy cases, there are very large shifts. The lowering of the energy is up to 1.6 eV in all functionals for the largest basis set, as shown in Table A1 in Appendix A. Thus, the 350 singlet excited states of EGCG are located within 4.20–10.72 at PBE0/6-31G(dp), while they are located within 4.25–9.16 at PBE0/6-311+G(d,p). The calculated absorbance spectra of EGCG in a water solvent at B3LYP/6-311+G(d,p), PBE0/6-31G(dp), and PBE0/6-311+G(d,p) using a 0.25 eV UV-vis peak half width at half height are depicted in Figure 2b along with the experimental spectrum. We observe that PBE0 is in better agreement with the experimental spectra and the PBE0/6-311+G(d,p) methodology predicts better the spectra of EGCG in water.

The main absorption bands for each of the local structure minima for the eight calculated isomers are presented in Table 3. The spectra are depicted in Figure 2b and Figure A1, Figure A2, Figure A3 and Figure A4 in Appendix A. The overall shape of the absorption spectra for the different isomers is the same. However, there are some differences between the two groups of isomers, i.e., the existence of the van der Waals H…O bond between the H atom of the pyrogallol ring and O of the gallate ring results in some shifts in the low-energy peaks. For example, the first excitation S0→S1, which corresponds to a HOMO→LUMO (H→L) excitation and is a CT state, shifts to lower energies by 20 nm for the second group of the isomers forming van der Waals (vdW) bonds. Similarly, the peak of the first group of the isomers (no vdW bond) at 264 nm is shifted to 256 nm in the second group of the isomers. On the contrary, the maximum peak at about 195 nm remains the same for both groups of isomers, as shown in Table 3. Overall, there is a very good agreement between the experimental spectrum and the computational ones, as shown in Figure 2. In Figure 2b, the experimental spectrum shows a shoulder peak at 228 nm. Computationally, this shoulder corresponds to a peak at 214 nm (see Table 1), which is blue-shifted about 14 nm and, thus, it is observed as a shoulder at the UV-vis absorption spectrum.

The PBE0/6-311+G(d,p) electron density plots of the frontier orbitals involved in the most important absorption excitations of the EGCG molecule in water are depicted in Figure 5 and Figure A3. In Table 1, the absorption bands and their respective assignments of the electronic transitions observed in the VUV spectra of EGCG in water are presented. The S1 excitation corresponds to a CT state from resorcinol and pyrogallol to gallate. The following three excitations present a similar character to that of S1. S5, which presents a high oscillator strength, corresponds to the n-π* transition involving oxygen electrons from gallic acid, i.e., the excited electron remains in the gallate group. At 200 nm, there are peaks with high oscillator strengths where the electron is located to resorcinol, gallate, or resorcinol groups. At 140 nm, partial CT states exist, where electron transfer is observed from resorcinol–pyrogallol groups to different smaller groups of the EGCG molecule.

Previous calculations using gallic acid [49] have shown that the explicit inclusion of water molecules affects the spectra of gallic acid, i.e., the explicit inclusion of 1–8 water molecules provided a correct picture for IR vibrations, but in the case of UV, showed a large red shift that did not reproduce the experimental findings and it was attributed to an overestimation of solvent effects. In this paper, to evaluate the importance of the inclusion of water, we added the solvent explicitly, i.e., in addition to considering water via its dielectric constant, we added a water molecule at the DFT level of theory, and we added seven and eleven water molecules at the ONIOM level of theory. Their absorption spectra for EGCG_7w (7 water molecules) and EGCG_11w (11 water molecules) are almost the same, showing that the results converged at the ONIOM level of theory. The main absorption maxima are presented in Table 4 and the calculated spectra are shown in Figure 6 and in Figure A3 and Figure A4 for the lowest two minima (a and e) from the two groups of minima. We observe that the inclusion of one water molecule results in a better agreement with the experimental spectra for the peak at 4.6 eV. However, the agreement with the experiment is not as good when seven or eleven water molecules are included, affecting the relative absorbance of the first and the second peaks. The oscillator strength of the second major emission is significantly larger when compared to the first one. Thus, our best result is obtained when one water molecule is added at the DFT level of theory. Regarding the two minima structures a and e and the relative positions of the two first major peaks at 267 and 203 nm, we observe that the ratio of the oscillator strengths of the peaks f203/f267 is 1.2(1.1) for a and 3.2(2.4) for e, without(with) the explicit inclusion of water at the PBE0/6-311G(d,p) level of theory, showing that the two different groups of minima present a different relative ratio of oscillator strengths. However, given that the population of the excitations in the area of 203 nm is very large, and with respect to the population of the excitations in the area of 267 nm, the difference in the shape of the absorbance spectrum is not very different for the two minima. Finally, it should be noted all three functionals, B3LYP, PBE0, and ωΒ97XD, combined with the 6-31G(d,p) and 6-311+G(d,p) basis sets, predict similar absorption peaks. The B3LYP and PBE0 predict almost the same λ values, while ωΒ97XD predicts blue-shifted values by about 10 nm compared to the B3LYP and PBE0 values. The inclusion of dispersion corrections via the ωΒ97XD functional results in more intense peaks (see f-values (oscillator strengths)) and a blue shift of the calculated absorption spectra. Among the functionals used, the PBE0 and B3LYP functionals are in better agreement with the experimental values; see Table 1 and Table 4.

It should be noted that, for the peak at 140 nm, the inclusion of water shifts the peak slightly from 134.7 (without the explicit inclusion of water) to 136.1 (one water) to 143.5 (seven water) to 144.6 nm (eleven water molecules); see Table 4. Moreover, it should be noted that water has a very small UV peak at 146.4 nm (see Figure 6a), which only slightly influences the spectrum of EGCG. Finally, regarding the small basis, the explicit inclusion of the solvent has the same effect as the large basis set and the peaks of the high energy are shifted to even larger energies (see Figure A3 and Figure A4).

### 2.3. Interaction of EGCG with DPPG Lipid

The VUV characterization of DPPG emulsions and thin films was carried out and the absorbance spectrum of DPPG thin films shows four bands at A) 138.2 ± 0.4, B) 145.8 ± 0.4, C) 169.8 ± 0.3, and D) 192 ± 2 nm [59]. These were assigned to: A) the π_C=O_→π^∗^_CO_ transition; B) to the phosphate group transitions; C) to several assignments as the second lone pair orbital on the carbonyl group oxygen, a transition in the carboxyl group, the promotion of an electron from the highest filled molecular orbital to an antibonding orbital of O-H, and water dissociation; D) either to the n′_O_→π^∗^_CO_ transition from the lone pair on the carbonyl oxygen to the antibonding π_CO_ valence orbital or to the valence shell electronic excitations of hydroxyl groups [55]. Therefore, as one would expect, the VUV spectrum of DPPG-EGCG is quite like the one of EGCG, as observed in Figure 7a, which presents the obtained VUV spectra of EGCG and DPPG-EGCG cast thin films prepared from aqueous solutions or emulsions.

Theoretically, the UV-vis spectra of EGCG in a water solvent with or without the explicit water molecule and with or without the presence of DPPG is shown in Figure 7b. Note that, in the calculation of the EGCG spectra in DPPG_EGCG and DPPG_EGCG_w, only the part of the spectra that belongs to the EGCG molecule is obtained. The spectrum that belongs to DPPG is plotted alone (pink line). Theoretically, the interaction of EGCG with lipids only slightly shifts its relative positions of the peaks, by about 6 nm, in agreement with the experimentally measured spectrum, and the explicit inclusion of one water molecule improves this agreement further regarding the relative absorbance of the peaks (see Figure 7b). Similarly, the spectra of the free DPPG and the interaction of DPPG with EGCG are similar, i.e., only small shifts up to 5 nm are observed (see Table 4), with the spectra of the interacting DPPG presenting a larger f-value than that of the free DPPG due to the interaction.

Both the experimental and calculated spectra of EGCG and of DPPG-EGCG have a peak around 150 nm, which is assigned to an electron transfer transition from resorcinol–pyrogallol groups to different smaller groups of the EGCG molecule. It was measured experimentally that the relative absorbance of the peaks changes mainly around this area and the peak around 150 nm becomes an intense one. Our calculated separated spectra of the DPPG part and EGCG part of the DPPG-EGCG, presented in Figure 7, show that the increase in absorbance, which is observed experimentally in the EGCG spectrum interacting with DPPG in this area, could be attributed both to the DPPG spectrum and the EGCG-DPPG interaction.

Regarding the two minima a (no vdW) and e (including vdW interaction) and the relative positions of two first major peaks at 267 and 203 nm, we observe that the ratio of the oscillator strengths of the peaks f203/f267 is 1.2(1.1) for a and 3.2(2.4) for e, without(with) the explicit inclusion of water at the PBE0/6-311G(d,p) level of theory, while the interaction with DPPG changes the ratio at 2.0(1.3) and 3.4(2.6). We observe that the explicit inclusion of water results in a similar ratio with and without the interaction of the lipid. The lipid results in a small increase in the ratio of the two peaks. Again, given that many excitations are located in the area of 203 nm, while only a few are in the area of 267 nm, the difference in the shape of the absorbance spectrum of the two minima is not very different. Finally, our results follow other experimental and computational methods [30,36], in which the EGCG molecules are deep inside the lipid bilayer, positioned below the lipid ester groups, generating a concentration-dependent lipid condensation.

## 3. Materials and Methods

### 3.1. Experimental Details

#### 3.1.1. Chemicals

Epigallocatechin-3-gallate (EGCG) (M.W. of 458.4 g/mol) and 1,2-dipalmitoyl-sn-glycero-3-phospho-(1′-rac-glycerol) (sodium salt) (DPPG) (M.W. of 744.96 g/mol) were purchased from MilliporeSigma (Burlington, MA, USA) and Avanti Polar Lipids (Alabaster, AL, USA), respectively, and their chemical structures are shown in Figure 1.

#### 3.1.2. Preparation of Liposomes Entrapping EGCG

Liposomes were prepared by dissolving phospholipids in a mixture of chloroform and methanol 4:1 (*v*:*v*) to achieve a concentration of 5 mM [37]. The chloroform and methanol molecules were then evaporated with a stream of nitrogen leaving the lipid molecules deposited on the walls of the falcon tube as a thin film. Ultrapure water (Millipore GmbH, Billerica, MA, USA) or a 450 μM EGCG aqueous solution was added to these lipid films hydrating them for a period of two hours. DPPG or DPPG-EGCG liposomes were obtained after sonication with a tip sonicator (UP50H, Hielscher Ultrasonics, GmbH, Teltow, Germany) of vesicle suspensions in an ice bath. The sonication procedure was repeated 15 times with 1 min intervals between sonication cycles of 30 s. The determination of the encapsulation efficiency of EGCG molecules followed the protocol presented in [39]. Briefly, DPPG+EGCG liposomes were submitted to dialysis using the membranes of regenerated cellulose (Spectra/Pro, Biotech, Minneapolis, MN, USA) with a cut-off size of 8–10 kDa to remove untrapped EGCG molecules. The calculated encapsulation efficiency of EGCG in DPPG liposomes was 67%.

#### 3.1.3. Thin Films

The films were deposited on calcium fluoride substrates for spectroscopy studies. The substrates were cleaned with a 2% Hellmanex aqueous solution (Hellma™, Müllheim, Germany) for 1 h and then rinsed exhaustively with pure water. Films of EGCG, DPPG, and DPPG-EGCG were obtained by casting the EGCG aqueous solution (450 μM) and the DPPG and DPPG-EGCG vesicles suspensions onto calcium fluoride disks. The water was removed by submitting the samples to primary vacuum during 12 h. All the films were prepared and characterized at room temperature.

#### 3.1.4. Characterization by Vacuum Ultraviolet (VUV) Spectroscopy

The high-resolution VUV photo-absorption spectra of the cast films were recorded at the ultraviolet beamline of the synchrotron radiation facility ASTRID2 [60] at Aarhus University, Denmark. The setup consists of a sample vacuum chamber containing up to three CaF_2_ sample disks and one reference disk mounted on an MDC SBLM-266-4 push–pull linear motion. The VUV beam light passed through the disks and the transmitted intensity was measured at 1.0 nm intervals using a photomultiplier detector (Electron Tubes Ltd., Uxbridge, UK). The transmitted light intensity and the synchrotron beam ring current were measured at each wavelength, with a typical resolution that is better than 0.08 nm. The sample chamber had MgF_2_ entrance and exit windows. The minimum wavelength was determined by the CaF_2_ substrates so that the lowest wavelength at which reliable data could be collected was ~125 nm. To avoid absorption from molecular oxygen in the air for wavelengths below 190 nm, a small gap between the sample chamber exit window and the photomultiplier detector was flushed with He gas. To calculate the absorbance, the light intensity spectra of a clean CaF_2_ disc/or quartz cuvette were measured before and after measuring the spectrum of the disc covered with the EGCG film.

The subtraction of the baseline was used to highlight the electronic transitions in the spectra. As the scattering contribution is rather strong from the sample films, a slight over-correction may result in apparently negative absorbance in small sections of the spectra. However, the importance of the scattering correction is not to correctly model the scattering in the experiment, but to make the transitions easily identifiable and comparable to theoretical calculations.

### 3.2. Theoretical Methods

The electronic structure of the EGCG molecule was calculated via density functional theory (DFT) and time-dependent DFT (TDDFT) calculations using the B3LYP [61,62], M06-2X [63,64], PBE0 [65,66,67], and ωΒ97XD [68] functionals and second-order perturbation theory, MP2, in conjunction with the 6-31G(d,p) [69] and 6-311+G(d,p) [69] basis sets. At first, conformational analyses were carried out using the B3LYP function to calculate the most stable minimum structure in the gas phase and water. The solvent was implicitly included employing the polarizable continuum model (PCM) [70]. In total, eight stable isomers were calculated; see Figure 3.

To find the low energy minima, geometry optimization calculations were carried out, including the solvent implicitly, explicitly, and both implicitly and explicitly. In addition, the interaction of the DPPG lipid with EGCG was calculated via the ONIOM method [71,72,73] in a water solvent with and without the implicit inclusion of water molecules. The system was defined as two regions (layers), with the high layer that was the EGCG calculated at the PBE0/6-31G(d,p) and PBE0/6-311+G(d,p) levels of theory and the low layer, which was the lipid calculated at the PM6 level of theory. Finally, the implicit inclusion of water molecules was performed either at the DFT level of theory or at the semi-empirical method, adding one, seven, and eleven water molecules. We decided to add eleven water molecules because the EGCG molecule has eleven oxygen atoms, and we added seven water molecules because, after the interaction of lipids with catechin, seven oxygen atoms were free to form van der Waals bonds with the water molecules; see Figure 4. The interaction energy between water molecules and EGCG was calculated, and it was corrected for the basis set superposition error (BSSE) using the counterpoise procedure [74,75], since such corrections are especially important for vdW systems [76].

The absorption spectrum of EGCG in the gas phase and water solvent implicitly, explicitly, and both implicitly and explicitly was calculated. In addition, the absorption spectrum of EGCG-DPPG species in a water solvent with and without the explicit inclusion of water was calculated. The effect of the implicit and explicit water solvent in the spectra was studied, as well as the interaction of the DPPG lipid. All calculated spectra were obtained through the inclusion of up to 350 singlet-spin excited electronic states, i.e., the spectra were in the range from 4 to 11 eV. It should be noted that the absorption spectrum of EGCG-DPPG was calculated via the ONIOM methodology, in order to isolate the spectrum of EGCG from the spectrum of the DPPG lipid. Finally, the DPPG spectrum was calculated.

Finally, it should be noted that the DFT and TD-DFT methodologies are appropriate for the calculation of the present systems. The EGCG, DPPG_EGCG, EGCG_xW, and DPPG_EGCG_xW systems are all closed shell, and a single-reference method, such as DFT, is appropriate both for geometry and energetics. Additionally, the DFT methodology contains part of both static and dynamic correlations, while methods such as the CASSCF methodology, which are necessary for the study of multireference systems such as radicals [51], do not treat the dynamic correlation [77]. However, for the accurate study of the vibration modes, CASSCF presents a good agreement with the experimental vibrational modes on similar systems [51]. The TD-DFT methodologies used has been used in similar systems and they can be regarded as adequate for the present study [78,79]. Calculations on caffeine molecules, including the first sphere of hydration, have shown that, for the calculation of the absorption spectrum, the inclusion of the hydration sphere is not necessary, while it is important for the emission spectra, which was not the case of the present study. Additionally, the use of the 6-31G(d,p) basis set led to a similar absorption spectrum to that with the use of the Def2TZVP basis set. Finally, the comparison of our computational absorption spectra with our experimental ones showed that the used methodologies were adequate for the present study.

All calculations were performed using the Gaussian 16 program package [80,81,82,83]. The coordinates of all the optimum structures are included in Appendix A.

## 4. Conclusions

The present study provides the first complete VUV photo-absorption spectrum of EGCG in the energy range from 4.0 to 9.0 eV, which was measured by high-resolution vacuum ultraviolet (VUV) spectroscopy and was calculated by DFT and TDDFT methodologies. Conformational analyses of EGCG were carried out, and the electronic structure and the absorption spectra of the lowest in energy structures were computed. The solvent was included both implicitly and explicitly, including a different number of solvent molecules. Additionally, the interaction of EGCG with the DPPG lipid was calculated and the effect of the EGCG–lipid interaction on the EGCG spectra was studied. Finally, for all experimentally measured absorption bands observed in the VUV spectra of EGCG in a water solvent, assignments to the calculated electronic transitions were provided.

Our theoretical calculated spectra are in good agreement with the experimental spectrum. Our best theoretical results were obtained at the PBE0/6-311+G(d,p) level, including the solvent both implicitly and explicitly. The interaction of EGCG with the lipid changes slightly the calculated spectra regarding the relative positions of the peaks, in agreement with the experimental findings.

Both spectra of EGCG and of DPPG-EGCG have a peak around 150 nm, which is assigned to an electron transfer transitions from resorcinol–pyrogallol groups to different smaller groups of the EGCG molecule. It was measured experimentally that the relative absorbance of the peaks changes mainly around this area and, moreover, the peak around 150 nm is an intense one. This increase in absorbance is attributed both to the DPPG absorbance and to the EGCG-DPPG interaction.

Considering the biological role of EGCG in protecting cellular membranes against UV radiation-induced damage, the present analysis of the electronic structure may have important implications for cancer therapy since the effectiveness of chemotherapeutic agents is higher in cancer cells when they are combined with EGCG.

## Figures and Tables

**Figure 1 ijms-26-01084-f001:**
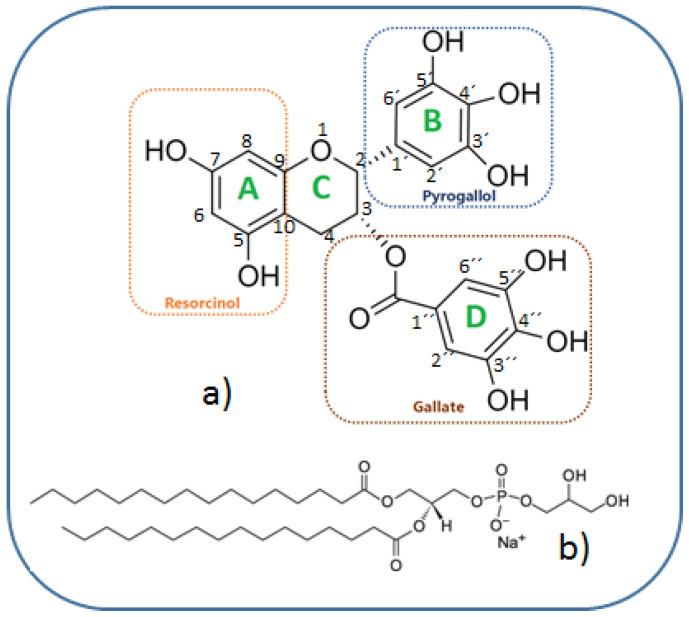
Chemical structures of: (**a**) EGCG and (**b**) DPPG.

**Figure 2 ijms-26-01084-f002:**
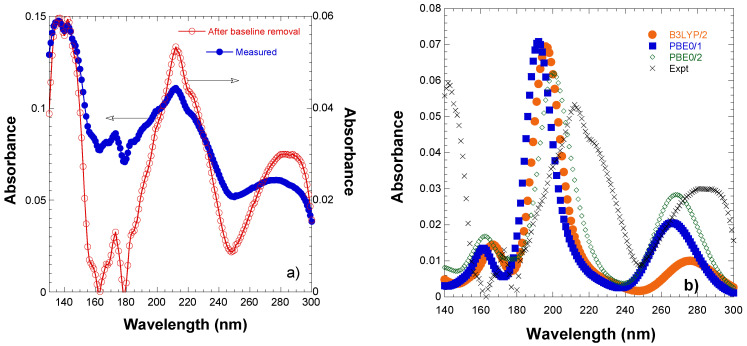
(**a**) Measured VUV absorption spectrum of EGCG cast film prepared from water solutions; (**b**) calculated absorption spectra of EGCG in water; 1: 6-31G(d,p), 2: 6-311+G(d,p).

**Figure 3 ijms-26-01084-f003:**
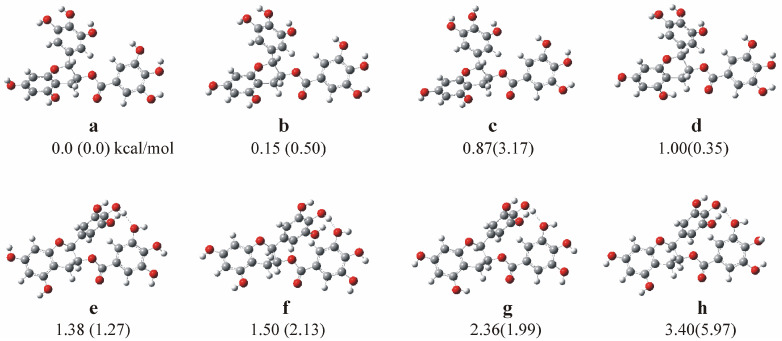
Calculated minima structures of EGCG. Relative energies (kcal/mol) in water (in the gas phase).

**Figure 4 ijms-26-01084-f004:**
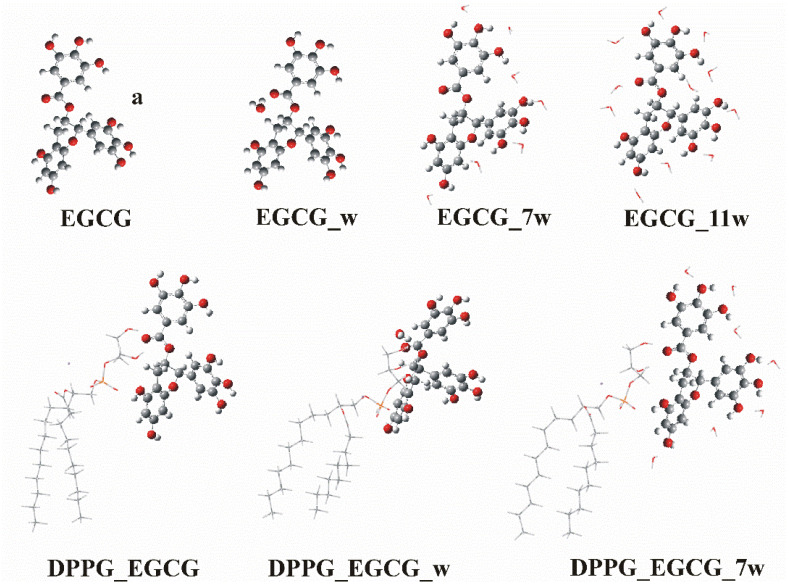
Lowest energy structures of EGCG, EGCG_w, EGCG_7w, EGCG_11w, DPPG_EGCG, DPPG_EGCG_w, and DPPG_EGCG_7w.

**Figure 5 ijms-26-01084-f005:**
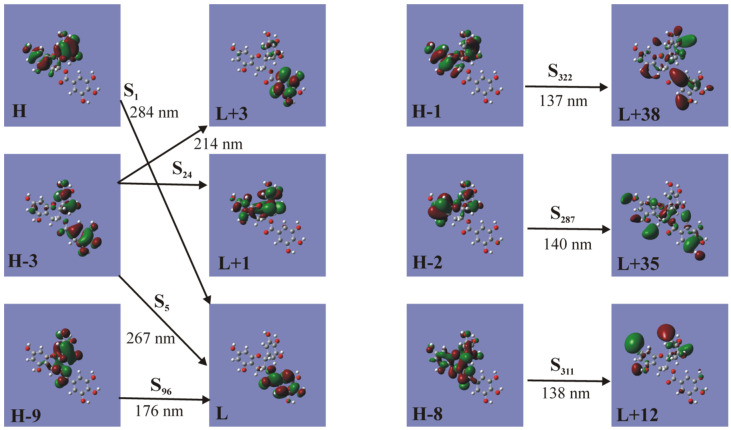
TPBE0/6-311+G(d,p) electron density plots of the frontier orbitals involving the most important absorption excitations of the EGCG molecule in water.

**Figure 6 ijms-26-01084-f006:**
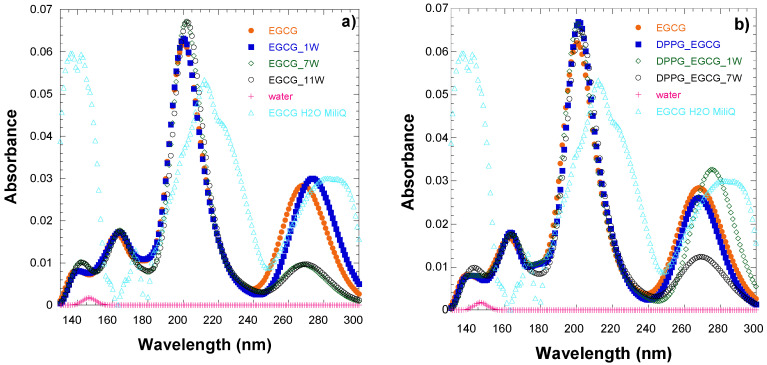
(**a**) Calculated UV–visible absorption spectrum of EGCG in water, explicitly including water molecules. (**b**) Calculated UV–visible absorption spectrum of EGCG within DPPG_EGCG in water, explicitly including water molecules.

**Figure 7 ijms-26-01084-f007:**
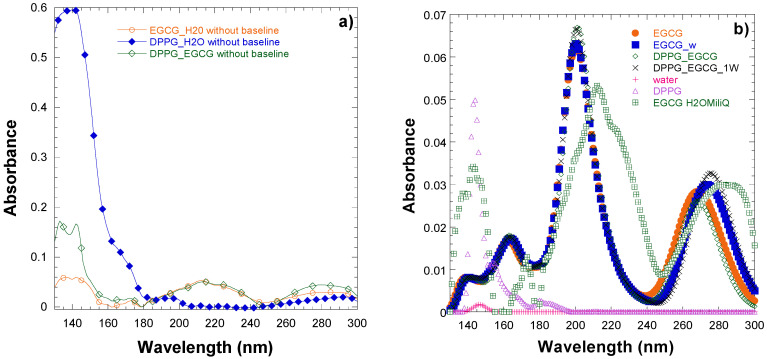
(**a**) Experimental UV–visible absorption spectrum of EGCG interacting with the DPPG lipid, (**b**) calculated UV–visible absorption spectrum of EGCG interacting with the DPPG lipid; EGCG_H2OMiliQ (expt).

**Table 1 ijms-26-01084-t001:** Position of the absorption bands in nm, full widths at half height (FWHW), and their respective assignments of the electronic transitions observed in the VUV spectra of EGCG in a water solvent. (Reported assignments are given in parentheses.)

Experimental Band Position (nm)	Calculated Band Position (nm)	Assignments to Electronic Transitions
140	135–140	Charge transfer states from resorcinol states with charge in gallate [51]
159	153	Transfer of charge mainly in pyrogallol and resorcinol
172	176,165	176 nm: Charge transfer (CT) states from pyrogallol to gallate; 165 nm: states with charge in gallate
212	200, 201, 201, 202, 202	No CT states: Charges in gallate or pyrogallol or resorcinol (on aromatic rings [37,56,57])
228	214	Charge is transferred to gallate
272	267	n-π* transition involving oxygen electrons from gallic acid
292	269	Charge transfer state from resorcinol–pyrogallol to gallate
315	279, 277, 284	279 nm: Charge transfer states from resorcinol–pyrogallol to gallate; 277 nm: States with charge in gallate; charge transfer states from resorcinol–pyrogallol to gallate (H→L)

**Table 2 ijms-26-01084-t002:** Relative energies *T_e_* (kcal/mol), van der Waals bond distances H…O (Å) between the H of the pyrogallol ring and O of gallate ring, and dihedral angles d (degrees) between resorcinol, gallate, and pyrogallol rings at the B3LYP, M06-2X, and PBE0/6-31G(d,p) (1) and 6-311+G(d,p) (2) levels of theory in water and in the gas phase.

		*T_e_*	H…O	d_RG_	d_RP_	d_GP_	T_e_
**B3LYP/1**	**a**	0.00		117.4	92.4	37.9	0.00
**B3LYP/2**				115.7	92.0	38.5	
**PBE0/1**				118.2	90.7	37.5	
**PBE0/2**				115.8	89.9	37.4	
**M06-2X/1**				125.2	87.3	41.6	
**M06-2X/2**				125.1	86.5	42.4	
**PBE0/2**	**e**	2.27	1.992	163.2	105.1	62.4	
**B3LYP/1**	**b**	0.15		117.3	92.2	37.9	0.50
**B3LYP/1**	**c**	0.87		117.8	93.4	36.0	3.17
**B3LYP/1**	**d**	1.00		118.1	92.0	38.3	0.35
**B3LYP/1**	**e**	1.38	1.987	153.9	97.1	59.8	1.27
**B3LYP/1**	**f**	1.50	1.987	155.3	98.1	60.3	2.13

**Table 3 ijms-26-01084-t003:** Absorption maxima, λ_max_ (nm), vertical excitation energies $∆E_{v}^{a}$(eV) S_0_→S_1_ absorption maxima corresponding to |H→L>, and f-values of the isomers of the EGCG molecule at different levels of theory in the gas phase and in water; 1: B3LYP, 2: M06-2X, 3: PBE0, i: 6-31G(d,p), ii: 6-311+G(d,p).

min		λ_max_	$∆E_{v}^{a}$	f	λ_max_	$∆E_{v}^{a}$	f	λ_max_	$∆E_{v}^{a}$	f	λ_max_	$∆E_{v}^{a}$	f
					**In the gas phase**								
**a**	**1i**	292.8	4.24	0.001	264.2	4.69	0.277	209.8	5.91	0.168	194.9	6.36	0.601
**b**	**1i**	293.1	4.23	0.0001	264.1	4.70	0.276	209.6	5.92	0.154	194.8	6.37	0.486
**c**	**1i**	297.7	4.16	0.002	263.7	4.70	0.279	209.9	5.91	0.123	195.6	6.34	0.381
**d**	**1i**	301.5	4.11	0.001	266.3	4.66	0.284	210.4	5.89	0.213	195.2	6.35	0.654
**e**	**1i**	312.5	3.97	0.002	256.3	4.84	0.233	204.6	6.06	0.080	194.6	6.37	0.420
**f**	**1i**	314.5	3.94	0.002	256.4	4.84	0.236	204.4	6.07	0.068	194.8	6.37	0.436
**g**	**1i**	320.2	3.87	0.003	258.6	4.79	0.254	207.8	5.97	0.060	197.7	6.27	0.528
**h**	**1i**	322.4	3.85	0.003	238.1	5.21	0.126	220.5	5.62	0.033	196.5	6.31	0.480
					**In the gas phase**								
**a**	**1i**	292.8	4.24	0.001	264.2	4.64	0.277	209.8	5.91	0.168	194.9	6.36	0.601
**a**	**1ii**	286.0	4.33	0.027	268.8	4.61	0.193	216.8	5.72	0.123	203.7	6.09	0.420
**a**	**2i**	250.6	4.95	0.019	236.8	5.24	0.311	194.1	6.39	0.458	185.2	6.70	0.624
**a**	**2ii**	256.5	4.83	0.023	242.5	5.11	0.214	215.8	5.74	0.054	203.4	6.10	0.421
**a**	**3i**	274.6	4.51	0.001	255.8	4.85	0.261	203.4	6.09	0.158	185.8	6.67	0.467
**a**	**3ii**	276.2	4.49	0.027	261.4	4.74	0.166	227.3	5.45	0.058	209.7	5.91	0.266
					**In the water solvent**								
**a**	**1i**	305.0	4.06	0.001	274.9	4.51	0.352	213.3	5.81	0.222	197.7	6.27	1.059
**a**	**1ii**	302.6	4.10	0.002	276.1	4.49	0.354	218.3	5.68	0.123	207.1	5.99	0.332
**a**	**2i**	253.6	4.89	0.025	244.3	5.07	0.419	197.7	6.27	0.416	187.8	6.60	1.252
**a**	**2ii**	257.4	4.82	0.027	246.9	5.02	0.399	217.7	5.70	0.054	202.6	6.12	0.429
**a**	**3i**	285.0	4.35	0.001	266.1	4.66	0.311	207.9	5.96	0.276	194.5	6.37	0.678
**a**	**3ii**	283.7	4.37	0.001	267.6	4.63	0.309	213.5	5.80	0.276	201.4	6.16	0.375
**e**	**3ii**	282.5	4.39	0.013	263.0	4.71	0.202	203.5	6.09	0.646	200.5	6.18	0.292

**Table 4 ijms-26-01084-t004:** Absorption maxima, λmax (nm), vertical excitation energies $∆E_{v}^{a}$(eV) S0→Sx absorption maxima, and f-values of the lowest energy a isomer of the EGCG molecule in a water solvent interacting with DPPG; 1: B3LYP, 2: PBE0, 3: ωΒ97XD, i: 6-31G(d,p), ii: 6-311+G(d,p).

	λ_max_	$∆E_{v}^{a}$	f	λ_max_	$∆E_{v}^{a}$	f	λ_max_	$∆E_{v}^{a}$	f	λ_max_	$∆E_{v}^{a}$	f	λ_max_	$∆E_{v}^{a}$	f
							**EGCG**								
**1i**	274.9	4.51	0.352	197.7	6.27	1.059	167.4	7.41	0.158	124.8	9.93	0.028	112.9	10.98	0.075
**1ii**	276.1	4.49	0.354	207.1	5.99	0.332	168.9	7.34	0.080	160.2	7.74	0.057	134.7	9.21	0.026
**2i**	266.1	4.66	0.311	194.5	6.37	0.678	161.7	7.67	0.188	128.1	9.68	0.051	109.3	11.35	0.049
**2ii**	267.6	4.63	0.309	201.4	6.16	0.375	175.7	7.06	0.084	164.6	7.53	0.077	134.7	9.20	0.028
**2ii** ^[a]^	263.0	4.71	0.202	203.5	6.09	0.646	169.8	7.30	0.070	159.2	7.79	0.139	134.9	9.19	0.026
							**EGCG_1w**								
**2i**	272.3	4.55	0.329	193.0	6.43	0.870	163.7	7.57	0.120	128.4	9.68	0.054	117.0	10.60	0.021
**2ii**	272.7	4.55	0.331	201.6	6.15	0.354	176.4	7.03	0.052	165.6	7.49	0.066	146.0	8.49	0.020
													(136.1	9.11	0.024)
**2ii** ^[a]^	267.8	4.63	0.248	203.6	6.09	0.596	167.4	7.41	0.091	159.6	7.77	0.129	141.0	8.79	0.015
							**EGCG_7w**								
**2i**	265.8	4.66	0.159	194.1	6.39	1.246	161.8	7.67	0.144	129.0	9.61	129.0	118.0	10.50	0.018
**2ii**	266.3	4.66	0.190	202.2	6.13	0.996	176.7	7.02	0.105	165.8	7.48	0.073	143.5	8.64	0.016
							**EGCG_11w**								
**2i**	265.1	4.68	0.262	194.1	6.39	0.824	161.5	7.68	0.197	129.0	9.61	0.089	120.7	10.27	0.024
**2ii**	268.2	4.62	0.204	202.9	6.11	1.002	177.9	6.97	0.102	164.6	7.53	0.075	144.6	8.57	0.016
							**DPPG**								
**2ii** ^[b]^	215.8	5.74	0.003	178.7	6.94	0.045	156.7	7.91	0.037						
**2ii** ^[c]^	211.0	5.88	0.002	180.2	6.88	0.029	155.4	7.98	0.025	142.4	8.71	0.040			
							**DPPG_EGCG**								
**2i**	265.5	4.67	0.353	194.0	6.39	0.776	161.7	7.67	0.171	130.1	9.53	0.040	119.8	10.35	0.025
**2ii**	268.6	4.62	0.197	201.9	6.14	0.396	178.3	6.95	0.087	165.9	7.47	0.076	140.2	8.85	0.022
**2ii** ^[a]^	263.3	4.71	0.199	204.0	6.08	0.673	170.6	7.27	0.087	160.2	7.74	0.176	142.1	8.73	0.027
	**DPPG_EGCG_1w**														
**2i**	272.2	4.56	0.375	192.9	6.43	0.906	163.6	7.58	0.221	128.1	9.68	0.038	124.1	9.99	0.033
**2ii**	274.6	4.52	0.364	202.8	6.11	0.480	179.9	6.89	0.135	165.7	7.48	0.134	139.9	8.86	0.018
**2ii** ^[a]^	269.2	4.61	0.257	204.1	6.07	0.659	169.7	7.31	0.079	160.8	7.71	0.113	138.1	8.98	0.016
**3ii**	251.2	4.93	0.357	197.5	6.28	1.328	156.3	7.93	0.089	153.4	8.08	0.122	123.6	10.03	0.036
	**DPPG_EGCG_7w**														
**2i**	264.8	4.68	0.251	194.0	6.39	1.256	161.3	7.69	0.190	129.0	9.61	0.065	121.4	10.21	0.037
**2ii**	268.2	4.62	0.227	202.4	6.13	0.910	178.1	6.96	0.122	164.9	7.52	0.077	141.3	8.77	0.018

^[a]^ minimum of EGCG. ^[b]^ DPPG in the DPPG_EGCG geometry. ^[c]^ Geometry-optimized D.

## Data Availability

The original contributions presented in the study are included in the article; further inquiries can be directed to the corresponding author. The raw data supporting the conclusions of this article will be made available by the authors upon request.

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
