# Peer review of "Electronic States of Epigallocatechin-3-Gallate in Water and in 1,2-dipalmitoyl-sn-glycero-3-phospho-(1′-rac-glycerol) (Sodium Salt) Liposomes"

_ijms, 2025, doi:10.3390/ijms26031084_

Round 1

Reviewer 1 Report

Comments and Suggestions for Authors

The manuscript presents interesting experimental and computational findings on the electronic structure and spectra of an important molecule. The authors may wish to adopt some of the following suggestions.

Major

Is TDDFT adequate for calculating the electronic structure? The molecule may well have significant multi-reference character. Reference 20 used CASSCF to address this, albeit on a fragment of the molecule. I appreciate that the size of the molecule may preclude a multi-reference treatment. Nevertheless, some discussion of this matter would be warranted.

In the TDDFT calculations was the Tamm-Dancoff approximation used?

Perhaps Table 3 would be better in the Supplementary Information.

The following paper perhaps warrants citing and discussion: Gier, H; Roth, W; Schumm, S; Gerhards, M. Structures of 1,2,3-trihydroxybenzene in the S0 and S1 states. J. Mol. Str., 610, 1-16. http://dx.doi.org/10.1016/S0022-2860(01)00786-4

Minor

Line 27. In the Abstract: “transitions transition” should be “transition”.
Line 43. The first sentence at the top of page 2 is repeatedly, albeit with different references at the end.
Line 57. “4.05D” requires a space before the “D”.
Line 58. Would it be useful to give the basis set in addition to the functional?
Line 91. Remove the hyphen from “hydro-xy”.
Table 1: “[Error! Bookmark not defined.]” requires attention.
Line 133. It is not clear what “350 excited states” refers to.
Lines 140, 156, 162, 170, 185, 325, 328, 375. “water solvent” should be “water”.
Line 145. “Walls” should be “Waals”.
Table 3. The basis sets should all be consistently described with “G”, not “g”.
Line 180. Is “17” a reference? If so, it needs brackets.
Lines 186, 187. “at” should be “at the”.

Author Response

Response to the Comments

The authors would like to thank the Reviewers for their positive feedback and useful comments, criticisms, and suggestions. The paper has been revised according to their suggestions, and we believe its quality has been considerably improved. We hope these changes are satisfactory.

All the answers to the comments raised by the Reviewer added below are marked in blue.

We run additional calculations at the MP2/6-311+G(d,p) and ωΒ97XD/6-311G+(d,p) levels of theory to calculate the interaction between water and EGCG molecules. Furthermore, the absorption spectrum of the EGCG-W system at the ωΒ97XD/6-311G+(d,p) method has been calculated.

Finally, for clarification reasons, the conformation analysis, geometries, and energetics that were given in section 3 (Materials and Methodology) have been moved to section 2 (lines: 117-169). This part has also been enriched with new results. The ordering of Tables, Figures, and references has been renumbered.

New references have been added, i.e., Refs. 25b, 40, 46, and 47, to reply to reviewers’ comments or due to the addition of new calculations (ωΒ97XD). Some typos have been corrected. All the changes introduced in the revised version of the manuscript are highlighted in yellow for a better perception of the Reviewers and the Editor.

Reviewer #1

The manuscript presents interesting experimental and computational findings on the electronic structure and spectra of an important molecule. The authors may wish to adopt some of the following suggestions.

Major

Q1: Is TDDFT adequate for calculating the electronic structure? The molecule may well have significant multi-reference character. Reference 20 used CASSCF to address this, albeit on a fragment of the molecule. I appreciate that the size of the molecule may preclude a multi-reference treatment. Nevertheless, some discussion of this matter would be warranted.

Author’s Reply: Yes, the used DFT and TDDFT methodology is adequate for the present systems. See text lines: 416-425.      

“Finally, it should be noted that DFT and TD-DFT methodologies are appropriate for the calculation of the present systems. The EGCG, DPPG_EGCG, EGCG_xW, and DPPG_EGCG_xW systems are all closed shell and a single reference method such as DFT is appropriate both for geometry and energetics. Additionally, DFT methodology contains part of both static and dynamic correlation, while methods such as the CASSCF methodology, which are necessary for the study of multireference systems such as radicals,[25] do not treat the dynamic correlation.[45] However, for the accurate study of the vibration modes, CASSCF presents good agreement with the experimental vibrational modes on similar systems. [25] The present used TD-DFT methodologies has been used in similar systems, and they can be regarded as adequate for the present study.[46]”

Additional calculations using the ωΒ97XD functional have been carried out, see Table 4 and text lines:158-165.

Reference 20 is in the revised manuscript, i.e., Ref 25, which has been enriched with ref 25b.

Three additional references have been added, Ref 40, 45 and 46.

Q2: In the TDDFT calculations was the Tamm-Dancoff approximation used?

Author’s Reply: No, the standard TDDFT calculations were carried out, line:385.

Q3: Perhaps Table 3 would be better in the Supplementary Information.

Author’s Reply:  The authors appreciate the reviewer's suggestion and adapted the manuscript accordingly.

Q4: The following paper perhaps warrants citing and discussion: Gier, H; Roth, W; Schumm, S; Gerhards, M. Structures of 1,2,3-trihydroxybenzene in the S0 and S1 states. J. Mol. Str., 610, 1-16. http://dx.doi.org/10.1016/S0022-2860(01)00786-4

Author’s Reply: This paper has added, new Ref 25b.

“… for the accurate study of the vibration modes, CASSCF presents good agreement with the experimental vibrational modes on similar systems. [25]”, lines 422-423

Minor

Line 27. In the Abstract: “transitions transition” should be “transition”.

Author’s Reply:  The text was altered accordingly.

Line 43. The first sentence at the top of page 2 is repeatedly, albeit with different references at the end.

Author’s Reply:  The reviewer is right to point out this typo. The sentence was rewritten to:

“Our previous studies revealed that the incorporation of EGCG protects the 1,2-dipalmitoyl-sn-glycero-3-phospho-(1'-rac-glycerol) (sodium salt) (DPPG) membrane, namely the phosphate and carbonyl groups of phospholipids, from oxidative stress induced by ultraviolet (UV) radiation [8-10].”, line 43.

Line 57. “4.05D” requires a space before the “D”.

Author’s Reply:  The text was altered accordingly.

Line 58. Would it be useful to give the basis set in addition to the functional?

Author’s Reply:  The authors agree. Please see line 53.

Line 91. Remove the hyphen from “hydro-xy”.

Author’s Reply:  The text was altered accordingly.

Table 1: “[Error! Bookmark not defined.]” requires attention.

Author’s Reply:  The text was altered accordingly.

Line 133. It is not clear what “350 excited states” refers to.

Author’s Reply:  It has been explained, line:184.

Lines 140, 156, 162, 170, 185, 325, 328, 375. “water solvent” should be “water”.

Author’s Reply:  The text was altered accordingly.

Line 145. “Walls” should be “Waals”.

Author’s Reply:  The text was altered accordingly.

Table 3. The basis sets should all be consistently described with “G”, not “g”.

Author’s Reply: The text was altered accordingly.

Line 180. Is “17” a reference? If so, it needs brackets.

Author’s Reply:  The text was altered accordingly.

Lines 186, 187. “at” should be “at the”.

Author’s Reply:  The text was altered accordingly.

Reviewer 2 Report

Comments and Suggestions for Authors

In this work, the authors investigated the spectroscopy of EGCG and its bonding with DPPG lipids using experimental vacuum ultraviolet (VUV) absorption measurements (4.0–9.0 eV) and DFT-based theoretical calculations. Experimental and theoretical data align well, with solvent inclusion improving the match. Electronic transitions observed in EGCG’s VUV spectrum in water are assigned through calculations. The DPPG-EGCG spectrum shows a strong peak near 150 nm, attributed to electron transfer between specific groups in the EGCG molecule, and the increased absorbance is linked to molecular interactions.

This work can be interesting for the computational chemistry and the industrial community. I would like to ask the authors to consider the minor comments below.

1. page 4, Figure 2

In the Figure 2b, the experiment spectrum shows a shoulder peak at ~225 nm. But this peak is missing in all calculated spectrums. Can the authors provide some discussions on this issue?

2. page 4, Table 1

In the second row, there is a typo: “[Error! Bookmark not defined.]” .

3. Are the results converged with respect to the number of water molecules? For example, more than 40 water molecules might be needed.

4. page 7, Figure 3

In addition to the electron density, it can be helpful to plot natural transition orbitals from TDDFT to show the electron transfer.

5. The long-range interaction can play an important role in the studied systems. Can the authors discuss the effects of not adding the dispersion correction in the DFT calculations?

6. For excited state modelling, 6-311+G(d,p) basis set might be too small. Typically aug-cc-pVTZ or equivalent basis sets are needed to provide converged results.

Author Response

Response to the Comments

The authors would like to thank the Reviewers for their positive feedback and useful comments, criticisms, and suggestions. The paper has been revised according to their suggestions, and we believe its quality has been considerably improved. We hope these changes are satisfactory.

All the answers to the comments raised by the Reviewer added below are marked in blue.

We run additional calculations at the MP2/6-311+G(d,p) and ωΒ97XD/6-311G+(d,p) levels of theory to calculate the interaction between water and EGCG molecules. Furthermore, the absorption spectrum of the EGCG-W system at the ωΒ97XD/6-311G+(d,p) method has been calculated.

Finally, for clarification reasons, the conformation analysis, geometries, and energetics that were given in section 3 (Materials and Methodology) have been moved to section 2 (lines: 117-169). This part has also been enriched with new results. The ordering of Tables, Figures, and references has been renumbered.

New references have been added, i.e., Refs. 25b, 40, 46, and 47, to reply to reviewers’ comments or due to the addition of new calculations (ωΒ97XD). Some typos have been corrected. All the changes introduced in the revised version of the manuscript are highlighted in yellow for a better perception of the Reviewers and the Editor.

Reviewer #2

In this work, the authors investigated the spectroscopy of EGCG and its bonding with DPPG lipids using experimental vacuum ultraviolet (VUV) absorption measurements (4.0–9.0 eV) and DFT-based theoretical calculations. Experimental and theoretical data align well, with solvent inclusion improving the match. Electronic transitions observed in EGCG’s VUV spectrum in water are assigned through calculations. The DPPG-EGCG spectrum shows a strong peak near 150 nm, attributed to electron transfer between specific groups in the EGCG molecule, and the increased absorbance is linked to molecular interactions.

This work can be interesting for the computational chemistry and the industrial community. I would like to ask the authors to consider the minor comments below.

  1. page 4, Figure 2

Q1: In the Figure 2b, the experiment spectrum shows a shoulder peak at ~225 nm. But this peak is missing in all calculated spectrums. Can the authors provide some discussions on this issue?

Author’s Reply: We thank the Reviewer, and it has been explained by:

Overall, there is a very good agreement between the experimental spectrum and the computational ones, see Figure 2. In Figure 2b, the experimental spectrum shows a shoulder peak at 228 nm. Computationally, this shoulder corresponds to a peak at 214 nm, see Table 1, it is blue shifted about 14 nm and thus it is observed as a shoulder at the UV-vis absorption spectrum.”, lines: 201-206.

Q2. page 4, Table 1

In the second row, there is a typo: “[Error! Bookmark not defined.]” .

Author’s Reply:  The text was altered accordingly.

Q3. Are the results converged with respect to the number of water molecules? For example, more than 40 water molecules might be needed.

Author’s Reply:  Yes, the results converged.

“Their absorption spectrum for EGCG_7w (7 water molecules) and EGCG_11w (11 water molecules) are almost the same, showing that the results converged at the ONIOM level of theory.”, lines 240-242.

“We have added eleven water molecules because the EGCG molecule has eleven oxygen atoms. We have added seven water molecules because, after the interaction of lipid with catechin, seven oxygen atoms were free to form van der Waals bonds with the water molecules, see Figure 4,” lines 124-127.

Q4: 4. page 7, Figure 3

In addition to the electron density, it can be helpful to plot natural transition orbitals from TDDFT to show the electron transfer.

Author’s Reply: For the main peaks, the coefficients of the molecular orbital transitions were large, or multiple orbitals had electron density in the same groups, showing that the natural transition orbitals will have the same electron density as the simple molecular orbitals.

Q5. The long-range interaction can play an important role in the studied systems. Can the authors discuss the effects of not adding the dispersion correction in the DFT calculations?

Author’s Reply: Additional calculations have been carried out using the ωΒ97XD/6-311G+(d,p) method to calculate the interaction between water and EGCG molecules, lines: 158-165

Furthermore, the absorption spectrum of the EGCG-W system at the ωΒ97XD/6-311G+(d,p) method has been calculated. See, Table 4 and text lines:258-265.

Q6. For excited state modelling, 6-311+G(d,p) basis set might be too small. Typically aug-cc-pVTZ or equivalent basis sets are needed to provide converged results.

Author’s Reply:  For the calculation of the absorption spectra 350 excited state have been computed at the B3LYP, PBE0, M06-2X and ωΒ97XD/6-311G+(d,p) levels of theory. Comparison of our present computational absorption spectra with our experimental ones, shows that the used basis set is adequate for the present study.

“The present used TD-DFT methodologies have been used in similar systems and they can be regarded as adequate for the present study.[47] Calculations on caffeine molecules including the first sphere of hydration have shown that for the calculation of the absorption spectrum, the inclusion of the hydration sphere was not necessary, while it was important for the emission spectra which is not the case of the present study. Additionally, the use of the 6-31G(d,p) basis set led to a similar absorption spectrum with the Def2TZVP basis set. Finally, a comparison of our present computational absorption spectra with our experimental ones, shows that the used methodologies are adequate for the present study.” lines: 424-432.